# Reactive HiTUS TiNbVTaZrHf-N_x_ Coatings: Structure, Composition and Mechanical Properties

**DOI:** 10.3390/ma16020563

**Published:** 2023-01-06

**Authors:** František Lofaj, Lenka Kvetková, Tomáš Roch, Jozef Dobrovodský, Vladimír Girman, Margita Kabátová, Matúš Beňo

**Affiliations:** 1Institute of Materials Research of the Slovak Academy of Sciences, 917 24 Trnava, Slovakia; 2Faculty of Mathematics, Physics and Informatics, Comenius University in Bratislava, 842 48 Bratislava, Slovakia; 3Advanced Technologies Research Institute, Faculty of Materials Science and Technology in Trnava, Slovak Technical University in Bratislava, 917 24 Trnava, Slovakia; 4Faculty of Science, Pavol Jozef Šafárik University in Košice, 040 01 Košice, Slovakia

**Keywords:** multicomponent nitride coatings, reactive high target utilization sputtering (HiTUS), hysteresis behavior, micromechanical properties, nanoindentation, structure–properties relations

## Abstract

High entropy metal sub-lattice stabilized nitride coatings based on multicomponent refractory transition metals (TM = Ti, Nb, V, Ta, Zr, Hf) are promising candidates for extreme conditions due to their high thermal, mechanical, and corrosion properties. The aims of the current work included the investigations of the possibilities of the novel High Target Utilization Sputtering (HiTUS) technique applied to reactive sputtering of TiNbVTaZrHf–xN coatings from the viewpoints of hysteresis behavior during reactive sputtering as well as the structure, composition, stoichiometry, and mechanical properties of the resulting coatings. With increasing nitrogen content, coating structures varied from amorphous in metallic alloy coatings to textured nano-columnar fcc structures. Despite certain deviations of TM from equiatomic concentrations, homogeneous solid solutions corresponding to single-phase multicomponent nitride analogous to high entropy stabilized compounds were obtained. Mechanical properties were found to be proportional to nitrogen content. The highest hardness H_IT_ ~ 33 GPa and indentation modulus E_IT_ ~ 400 GPa were found in a slightly sub-stoichiometric (~42 at% nitrogen) composition. H_IT_/E_IT_ and limited pillar split measurements suggested that these coatings exhibit low fracture toughness (around 1 MPa.m^1/2^). The work confirmed that reactive HiTUS is suitable for the preparation of multicomponent nitrides with the control of their stoichiometry and mechanical properties only via nitrogen additions.

## 1. Introduction

Strong nitride formers among transition metals (TM) including Ti, Zr, Hf, V, Nb, and Ta, form single nitrides with high thermal stability combined with low thermal conductivity, high hardness, and corrosion resistance [1,2]. Additional control and enhancement of these properties via simultaneous alloying by several refractory transition metals were greatly promoted by the introduction of the concept of so-called “high entropy alloys” (HEA) by Cantor et al. [3] and Yeh [4] in 2004. The main idea of HEA is that homogeneous solid solutions with fcc (face-centered cubic), bcc (body-centered cubic), or even hcp (hexagonal close-packed) structures consisting of at least five equimolar metals can be stabilized by configurational entropy. Additional stabilizing “core” effects involve lattice distortion due to variations of atomic radii of different metals, sluggish diffusion resulting from diffusion barriers in such distorted lattice, and the possibility of a synergistic “cocktail” effect [5,6]. The concept of bulk metallic HEA paved the path for the development of high entropy ceramics (HEC) via the introduction of boron, carbon, nitrogen, or oxygen [7,8,9,10,11]. In the case of TM-based nitride ceramics, corresponding transition metals were classified depending on their affinity to nitrogen. Elements with high formation enthalpies from groups (3-) 4–5 groups of the periodic table ((Y) Ti, V, Zr, Nb, Hf, Ta) correspond to strong nitride formers. The metals from 6–10 groups (Cr, Mn, Fe, Co, Ni, Mo, Ru but also Al and Si) with lower formation enthalpies belong to weak nitride formers and even to no-nitride formers (e.g., Cu) [12]. Strong nitride formers easily form interstitial solid solutions in which nitrogen atoms fill voids in the metal sub-lattice, typically with a NaCl-type crystal structure. It involves a mix of covalent, metallic, and ionic bonding resulting in higher mechanical and thermal properties than in single TM-N compounds [7,13]. However, high configurational entropy in the multicomponent nitrides applies only to the metal sub-lattice, whereas the nitrogen sub-lattice is practically not affected. Thus, high entropy ceramics, including nitrides, should be more accurately described as “materials with high entropy metal sub-lattice” [14,15].

The recent reviews on high entropy ceramic coatings can be found in [12,16,17]. Probably the first multicomponent high entropy-like nitride coatings were deposited in 2004 from FeCoNiCrCuAl (Mn) targets by reactive DC magnetron sputtering (DCMS) [18]. The metals involved weak nitride formers, which caused the near-stoichiometric nitrides to be amorphous. Subsequent studies were focused on the nitrides containing various combinations of AlCr-Nb/Si/Fe/Mo/Ni/Ta/Y/Zr/V [12]. The first nitride coatings involving four strong nitride formers—Hf, Ti, Zr, and V combined with Cr—were prepared in 2011 also by reactive magnetron sputtering [19]. Their structure varied from amorphous without nitrogen to textured fcc solid solutions in (near-)stoichiometric compositions obtained at sufficiently high nitrogen contents. The highest hardness of 23.8 ± 0.8 GPa and modulus 267 ± 4 GPa were reached in the (111) textured coating saturated with nitrogen. In 2012–2013, nitride coatings involving five (Ti-V-Zr-Nb-Hf and Ti-Zr-Nb-Hf-Ta) strong nitride formers were deposited by arc [9,20] and reactive magnetron co-sputtering, respectively [10,21]. Hard textured (111) fcc solid solutions formed in all cases. The first nitride coatings involving six strong nitride formers (Ti-V-Zr-Nb-Hf-Ta) were produced by arc in 2014 [22] and 2015 [23]. In (111) textured fcc near-stoichiometric coatings, hardness values in the 36–51 GPa range were obtained [22,23].

The application of sputtering techniques different from arc and DCMS that provide higher levels of ionization of the sputtered species, e.g., High Power Impulse Magnetron Sputtering (HiPIMS), to multicomponent nitride coatings is limited to a few studies up to now, mostly on AlCrTiVZr-N [24,25] and TiZrNbTFe-N systems [26]. The HiPIMS-made coatings were denser and exhibited higher hardness (41.8 GPa) than those made by DCMS (36.2 GPa). Another novel sputtering technique, High Target Magnetron Sputtering (HiTUS), was originally developed for oxide coatings for optical applications [27,28]. The main difference between HiTUS and magnetron sputtering is plasma decoupling from the target (cathode) and plasma generation in an independent RF plasma source. It results in the absence of a racetrack and sputtering from practically the entire surface of the target, which gave the technique its name. In the case of reactive processes, it causes low sensitivity to target poisoning which is reflected in the suppression of hysteresis behavior. HiTUS had already been successfully applied to the deposition of hard W-C:H coatings using (reactive) hybrid PVD-PECVD processes based on acetylene or methane additions [29,30,31,32]. The obtained hardness values were equal or often slightly higher than in the case of DCMS and almost as high as in the analogous HiPIMS-made W-C:H coatings [29,30,31,32]. However, HiTUS has not been applied to the sputtering of multicomponent metallic coatings analogous to HEA nor to the reactive sputtering in analogous nitride systems. Its potential compared to DCMS coatings is also not known. Therefore, this work investigates various aspects of reactive HiTUS in the deposition of hard multicomponent TM-based nitride coatings.

## 2. Experimental Procedure

### 2.1. Coating Deposition

The studied TiNbVTaZrHf-xN coatings were prepared by High Target Utilization Sputtering (model S500, Plasma Quest Ltd., Hook, UK) from a TiNbVTaZrHf target with a diameter of 76.2 mm in an Ar atmosphere with variable nitrogen flows. The nominal composition of the commercially obtained target (Testbourne, Whitchurch, UK) was Ti—20 at%, Nb—18 at%, V—20 at%, Ta—18 at%, Zr—12 at%, and Hf—12 at%. The true average concentrations of the elements in the target measured by EDS are indicated in Table 1. The maximum differences between the nominal composition and true target composition were up to +6.4 at% in V and −5.0 at% in Ta; the deviations in the other TM elements were below 2 at%.

The substrates involved polished (0001) sapphire and (111) Si wafers, which were ultrasonically cleaned in acetone and by plasma just prior to the onset of deposition. The coatings deposited on Si substrates were used for thickness measurements on the fractured cross-sections. To ensure good coating adhesion, gradient TiNbVTaZrHf-xN bond layers were applied. Gradient composition was obtained by a gradual increase of nitrogen flow additions into the sputtering atmosphere with 1 sccm and 1 min step starting from x = 0 sccm up to a nitrogen flow in top coating. Based on the deposition rate of ~20 nm/min, the bond layer thicknesses varied from 0 nm up to around 200 nm in the coatings produced with the highest nitrogen additions. Despite the importance of bond layers for coating adhesion, their role in structure formation and mechanical properties was not considered.

The common deposition parameters used for all coatings were: RF power on the remote plasma source–1800 W; RF power on the target—700 W; RF bias on the substrate—5 W (corresponding to DC bias of around −45 V); substrate temperature −300 °C; base pressure < 7.5 × 10^−3^ Pa; and initial working pressure ~0.76 Pa. This working pressure was obtained in an around 80 dm^3^ vacuum chamber with a turbomolecular pump with a pumping speed of 1250 L/s at a constant Ar flow of 120 sccm. The target–substrate distance was 17 cm. The only variable of the reactive HiTUS included additions of nitrogen in the range from 0 sccm up to 10 sccm in 2 sccm steps. Nitrogen additions resulted in an increase in the working pressure up to 0.82 Pa. The constant deposition time of 90 min resulted in the coatings with thicknesses in the range of 1.2–1.7 µm.

### 2.2. Structure, Mechanical Properties, and Thermal Stability

The basic structure observations were performed on the fractured cross-sections of the coatings on Si substrates using scanning electron microscopy (Auriga Compact and EVO MA15, Zeiss, Germany). Their chemical compositions were primarily measured by Energy Dispersive Spectroscopy (EDS) with an SDD detector (Oxford Instruments, 80 mm^2^, Abingdon, UK) from “point” and “area” measurements, with the contributions from the substrate intentionally excluded from the calculations.

Due to the limitations of quantitative measurements of nitrogen (and other light elements) in EDS, a combination of Ion Beam Analysis (IBA) methods was used for the depth profiling of nitrogen and TM elements concentrations. The measurements were performed on a 6 MV tandem ion accelerator (Tandetron, HVE, Groningen, The Netherlands) and IBA end station [33,34] at the Advanced Technology Research Institute (ATRI) of the Faculty of Materials Science and Technology in Trnava, Slovak University of Technology. The employed IBA methods involved Rutherford and Elastic (non-Rutherford) Backscattering Spectrometry (RBS/EBS), Nuclear Reaction Analysis (NRA), and Particle-Induced X-Ray Emission Spectrometry (PIXE) [35]. The energy of the primary He^+^ beam with a cross-section of 2.5 × 2.5 mm was 5.65 MeV to ensure the measurement of the depth concentration profile of all present elements over the entire coating thickness with optimum sensitivity and accuracy.

Although RBS analysis is considered a quantitative method without the use of reference samples [36], it was practically impossible to distinguish pairs of adjacent TM elements such as Ti/V, Zr/Nb, and Hf/Ta in the coating. PIXE analysis was therefore used to determine the ratio between adjacent transition metals and subsequently to calculate the contribution of each element in the pair. Nitrogen depth profiles were determined from nuclear reaction ^14^N(^4^He,p_0_)^17^O for the detector angle of 165° to the primary beam. The backscattered alpha particles were absorbed by a 36 nm thick Kapton absorber foil in front of the detector.

Simulation SIMNRA ver. 7.03 [37] (for RBS, EBS, NRA) and fitting GUPIXWIN [38] (for PIXE) programs were used to evaluate the measured energy spectra and to determine the depth concentration profiles of individual elements. The elastic backscattering differential cross-section data of ^12^C((α,α_0_)^12^C; ^14^N((α,α_0_)^14^N; ^16^O((α,α_0_)^16^O for SIMNRA simulations were taken from SigmaCalc 2.0 [39] and the cross-section data for the nuclear reaction ^14^N((α,p_0_)^17^O at theta 165° from [40].

The crystal structure of the coatings was analyzed using X-ray diffraction (PANalytical X’Pert Pro) with CuK_α_ radiation in a symmetrical Bragg–Brentano (B-B) setup with θ/2θ-scanning and in grazing incidence (GI) setup with detector 2θ-scanning at the fixed incidence angle of 1.5°. In symmetric B-B scanning, the diffraction vector was slightly offset by 2° from the strong substrate reflections. GI setup results in a much longer optical path and signal intensities from the coating than in B-B. The advantage of GI is that it reveals the crystal planes approximately parallel to the coating surface. The resulting GI and B-B diffractograms were obtained by averaging three measurements at three various azimuthal sample positions for improved data statistics. The crystallite size (CS) was calculated from the average width of the peaks using the Williamson-Hall method. It was also used for the determination of microstrains (ε).

The details of coating structures were investigated by high-resolution transmission electron microscopy (TEM) (model JEM 2100F, Jeol, Japan) on thin foils. The foils were prepared from two as-deposited coatings glued together using a standard procedure involving polishing, dimpling, and ion milling.

The measurements of mechanical properties of all coatings were based on nanoindentation with diamond Berkovich tip (G200, Agilent, Santa Clara, CA, USA) and the Continuous Stiffness Method for Thin Films (CSMTF) method [41]. According to the recommendations in [42], the tests were performed in constant strain rate mode (0.05 s^−1^) with the amplitude of the sinusoidal signal of 2 nm and frequency of 45 Hz. The measurements were performed on two sets of 16 indents up to a predefined depth of 800 nm in two locations approximately 1 mm apart on each coating. The measurement yielded two hardness and two indentation modulus depth profiles that were averaged from at least 2 × 10 valid tests. The mechanical properties of the coatings without the influence of substrate were determined from the maximum and/or plateau on the corresponding profiles extrapolated to zero depth (load). The depth range for the extrapolation was from >80 nm (given by the tip surface area calibration) up to around 200–300 nm, i.e., within or even above the maximum depth limit corresponding to the 10% rule [43]. Such a high upper limit for extrapolation resulted from improved elimination of substrate influence in the CSMTF method. The inputs required for calculating the indentation modulus in CSMTF include coating thickness, Young’s modulus, and Poisson’s ratio of the substrate. The thicknesses were measured by scanning electron microscopy (SEM) on the fractured cross-sections of the coatings deposited on Si wafers. The Young’s moduli of Si and sapphire substrates used in the calculations were 187 GPa and 435 GPa, and the corresponding Poisson’s ratios were 0.223 and 0.29, respectively. The effects of the bond layer and possible residual stresses on indentation hardness and modulus values were not considered.

## 3. Results

Hysteresis behavior during reactive magnetron sputtering relates to the effects of target poisoning on current, voltage, and working pressure at different flows of reactive gas [44,45]. Therefore, the possibilities of target poisoning and hysteresis behavior in HiTUS were investigated at first to understand the differences between conventional and reactive sputtering in HiTUS and to optimize the deposition conditions.

### 3.1. Plasma Conditions during Reactive HiTUS

Decoupling of the target from plasma discharge in HiTUS causes the voltage and current at the remote plasma source not to be influenced by reactive gas. The only easily detectable indication of the reactive processes would be the pressure changes [32]. Figure 1a shows the evolution of the total working pressure *p*_0_ (open circles) in the chamber with the plasma generated by the remote plasma source but without sputtering (RF power on the target was off). The pressure p_0_ measured by baratron changed linearly with the increase and/or decrease of nitrogen flow, and no differences were observed during the whole cycle. The small full symbols correspond to working pressure *p* during three cycles of nitrogen flow increase and decrease after RF 700 W was applied to the target and sputtering occurred. Immediately after the onset of sputtering at 0 sccm N_2_, *p* was systematically increased from 0.75 to 0.765 Pa. Such an increase cannot be related to reactive sputtering because of the absence of nitrogen. It was therefore attributed to plasma heating after the application (addition of energy) of sputtering and subsequent Ar expansion. Thus, the *p* increase at 0 sccm N_2_ was considered an artifact and was subtracted from the experimental values for further consideration. Three independent cycles of *p* measurements in Figure 1a slightly deviated from each other. To reduce the influence of the measurement errors, these three measurements were averaged, and only the average *p^av^ =* [*p*^(1)^ + *p*^(2)^
*+ p*^(3)^]/3 values were used. Figure 1b shows the curves of the average pressure difference ∆*p^av^ = p^av^ − p*_0_ during the nitrogen flow increase/decrease cycle. The physical meaning of ∆*p^av^* is related to the amount of reactive gas consumed by the reaction with the sputtered material leading to nitride formation [46]. Figure 1b shows ∆*p^av^* dependence after the elimination of the temperature artifact. Despite the relatively large scatter, all corrected ∆*p^av^* values were negative. The curve connecting average points during an increase in N_2_ flow decreased up to around 8 sccm N_2_, implying full nitrogen consumption. In the 8–12 sccm nitrogen flow range, ∆*p^av^* and nitrogen consumption remained approximately constant. At flows > 12 sccm, it increased and then stabilized at values slightly higher than the minimum. When the nitrogen flow decreased in the second part of the cycle, a very similar curve simply shifted upward by about 0.015 Pa. This shift was barely larger than the accuracy of the measurement. The hysteresis behavior, in this case, would therefore be very small and close to the error of measurement.

Despite almost negligible hysteresis behavior, the conclusion from Figure 1b that is important for reactive HiTUS deposition optimization is the existence of at least two regimes:full consumption of nitrogen at nitrogen flows below 8 sccm;decrease and saturation of nitrogen consumption at flows above 8 sccm.

### 3.2. Deposition, Structure, Phase, and Chemical Composition of Coatings

#### 3.2.1. Deposition and Deposition Rates

Table 2 overviews the studied HiTUS coatings deposited at several nitrogen flow levels. One additional coating without nitrogen and substrate temperature of 500 °C and another one with 6 sccm nitrogen at 300 °C but with RF 50 W bias (corresponding to DC bias of around −295 V) were added to explore the effects of substrate temperature and bias, respectively. Table 2 also contains coating thicknesses visible on fracture surfaces ranging from 1.2 to 1.7 µm. The average deposition rates calculated from these thicknesses are plotted in Figure 2 in dependence on nitrogen concentration taken from later IBA measurements and nitrogen flow.

The deposition rates followed two different dependences in two nitrogen flow ranges and two nitrogen concentration ranges. From 0 sccm up to around 6 sccm N_2_ resulting in nitrogen concentrations from 0 to 45.5 at. %, the rates decreased insignificantly from 0.316 nm/s (19 nm/min) to around 0.29–0.300 nm/s (17.4–18 nm/min). Rates dropped dramatically above 6 sccm and at stoichiometric 50 at% nitrogen concentrations. The reason for the existence of both ranges can be elucidated from their combination with the two nitrogen flow ranges related to their consumption mentioned in Figure 1b.

The increase of the substrate temperature to 500 °C in the coating without nitrogen and the bias in the case with 6 sccm N_2_ resulted in a decrease in deposition rates. The reasons may be related to the enhanced scattering of sputtered species in Ar atmosphere at higher temperatures. RF bias of 50 W corresponding to almost −300 V in the DC case was comparable to the values applied to the target during DC sputtering. Thus, the reduction of the deposition rate could be attributed to re-sputtering from the growing coating.

#### 3.2.2. Structure and Chemical Composition

Figure 3 shows fracture surfaces obtained on HiTUS-made coatings. In the coatings deposited with 0 sccm and 2 sccm N_2_ additions, the fracture surfaces were flat and featureless, indicating a brittle fracture and suggesting an amorphous or nanocrystalline structure. On the fracture surfaces of the coatings produced with ≥6 sccm N_2_ additions, columnar grains implying oriented textured crystalline structures were observed. The topography of the fracture surface in the coating obtained with 4 sccm N_2_ was also rough, but the columnar grains were not so clear. Fracture topography in this coating can be considered to be a result of a transition from fully brittle to intergranular fracture in crystalline textured structures.

High-resolution TEM (HRTEM) studies on thin foils confirmed the amorphous structure of the coating produced with 0 sccm N_2_ (Figure 4). It agreed with broad featureless rings in the selected area electron diffraction (SAED) patterns (see insert in Figure 4). The corresponding azimuthal integral SAED also indicated an amorphous structure. However, small (~1 nm) crystallites were also occasionally present. They could be assigned to crystallization nuclei, suggesting an early stage of localized crystallization.

TEM observations in the coating deposited with 2 sccm N_2_ showed a crystalline-like structure consisting of columnar grains growing perpendicularly to the substrate surface (Figure 5a). Their diameters were in the 10 nm range, and they were embedded in a residual amorphous phase. At higher magnifications, the columns were found to consist of many nanocrystallite domains with distinct preferred orientations and oriented mainly perpendicular to the substrate surface (Figure 5b). Very intense SAED reflections confirmed the presence of preferred orientation in a just-forming fcc structure. The spots’ broadening of ~18° in an angular direction was attributed to misorientation from preferred directions. The diffraction spots in the first circle correspond to 111 reflections. Next to them, a second ring of weaker 200 reflections can be observed. The existence of both 111 and 200 is strongly supported by the right shoulder in the most intense first peak of the azimuthal integral of the SAED pattern. The two strongest spots, overlapping 111 and 200, are oriented in a direction perpendicular to the substrate surface.

Figure 5c shows the image cut-off filtered in frequency space, which displays more details within these domains. The domains were full of dislocation-like defects resulting in their small misorientation, which made their structure regular only in a very small size range. The dotted lines in Figure 5c were introduced as a guide for eyes to indicate possible small-angle or even twin-like boundaries between such nanocrystalline domains. Thus, the structure of 6TM-2N coating could be described as an early stage of crystallization from an amorphous matrix with an already formed and growing but still only partially developed texture in the fcc lattice.

At 6 sccm nitrogen, well-defined crystalline columns with two orientations slightly declining from the normal direction to the substrate surface were identified (see arrows in Figure 6a), besides some amorphous phases among them. The diameter of the columns was in the range of 30–40 nm (Figure 6b). HRTEM (Figure 6c) confirmed the presence of two preferred orientations in the columns: (200) and simultaneously (111) planes close to parallel with the sample surface. The size of crystalline domains was much larger, and there were far fewer packing defects than in Figure 5c. Both SAED and its azimuthal integral showed the presence of much more focused, distinct, and intensive peaks in the first two rings corresponding to diffractions from (111) and (200) planes which was in full consent with the results in Figure 6c. The lattice parameter calculated from SAED was 0.4496 nm.

EDS mapping was subsequently performed on the same thin foils in STEM mode. Figure 7 visualizes TM distributions in the alloy coating deposited with 0 sccm N_2_. When a possible influence of thickness reduction toward the edge is considered, the distributions of all TM elements seem to be homogeneous.

Semi-quantitative atomic concentrations of individual TM elements in a thin foil of TiNbVTaZrHf-0N coating were obtained by averaging the data from seven different areas (Table 3). They varied from ~32 at% of Ti to ~5.5 at% of Hf. These variations indicated substantial deviations of individual TM concentrations from ideal equiatomic ~16.7 at% concentration in 6-component alloys. A relatively large scatter of TM element concentrations simultaneously indicated that the alloy is not fully homogeneous in the <100 nm range. Additional measurements were therefore performed using the top view in SEM from four areas with a much larger size (900 × 680 µm^2^) to obtain more representative average values. Then, the deviations from the equiatomic concentration were in the range from +2.3 at% (Nb) to +4.2 at% (Ti), whereas the deficits of −6.8 at% and −5.4 at% were observed in the cases of Zr and Hf, respectively. Despite these macro- and nano-scale deviations from an equiatomic value, the obtained HT-6TM-0N coating can still be considered a relatively homogeneous single-phase solid solution.

Table 3 also allows you to compare the TM concentrations in the coating to those in the target. Excess concentrations in the coating indicated enhanced sputtering of Ta (+6.7 at%) and Ti (+2.4 at%), whereas less sputtering occurred in V (−7.1 at%). Other elements had deviations of <1 at%. Thus, additional adjustment of the concentration of each TM in the target would be necessary to approach equiatomic TM concentrations in the coatings.

Figure 8 shows analogous EDS maps in thin foil made from HT-6TM-2N coating. Similarly, as in the metallic alloy coating, not only all TM elements but also nitrogen was distributed approximately homogeneously at this scale. Nitrogen concentration in thin foil was above 40 at%, whereas only 29 at% was found on the fracture surface by EDS/SEM.

EDS mapping applied to the 6TM-6N coating produced distributions practically identical to those in Figure 8 and confirmed the presence of a homogeneous solid solution. Local variations of element concentrations from thin foil, fracture surface, and average concentrations from large top view areas are compared in Table 4. The “nano-scale” nitrogen concentrations obtained by EDS in TEM scattered from 42 at% up to 55 at% with a mean value of 48 at%. It was the same as the nitrogen concentration obtained on a rough fracture surface from EDS/SEM. However, the representative average nitrogen concentration from the top view EDS/SEM measurements exceeded 55 at%.

EDS concentrations of TM elements obtained in TEM and SEM deviated by −6.5 at% (Ti) up to +4.3 at% (Ta). At the same time, depending on the measurement configuration, their deviations from an ideal equiatomic concentration of 8.3 at% (in stoichiometric 6-TM-50 at% N composition) varied. The smallest differences (+2.8 at% in Ti and −3.6 at% in Hf) were obtained in top view SEM, which was in the same range as in the metallic alloy coating.

The average concentrations of nitrogen and TM elements from top view EDS/SEM measurements for all coatings were summarized in Table 5 and Figure 9. Nitrogen concentrations increased rapidly from 0 at% to ~55 at% at 4 sccm N_2_. They saturated and remained within the 50–57 at% range at higher nitrogen flows. When the uncertainty due to the partial overlap of N and Ti peaks in EDS spectra was taken into account, nitrogen concentrations obtained at >4 sccm N_2_ flows could be considered approximately constant and near stoichiometric.

TM element concentrations decreased proportionally to nitrogen concentrations. There were also deviations from an ideal equiatomic concentration of 8.3 at%. Those of Ti (>10 at%) were approximately twice higher than those of Zr and Hf (5.0 and 5.5 at%, respectively), but the concentrations of V, Ta, and Nb were close to the ideal concentration.

To verify the concentrations obtained from EDS measurements, a combination of IBA methods (RBS-EBS-NRA-PIXE) was used as a reference because RBS is considered to be an absolute analytical method. Despite that, according to a very conservative estimation, the general uncertainty of TM concentrations determined by IBA was <5 at%, and that of nitrogen < 10 at%, IBA results were considered more reliable.

The concentrations from IBA measurements were compared with the earlier results from EDS in Table 5. The corresponding plot in Figure 10 shows that the coatings produced with up to 6 sccm N_2_ were sub-stoichiometric, and stoichiometry was achieved at 8 sccm and 10 sccm N_2_ flows. Nitrogen oversaturation implied from EDS measurements (see Figure 9) above 4 sccm N_2_ was not confirmed. In the TM sub-lattice, tendencies remained the same as in Figure 9, but the spread of V, Ta, and Nb concentrations was larger. Bias in the coating with 6 sccm N_2_ caused only a small (−1.6 at%) reduction in nitrogen content.

Figure 11 illustrates the differences between EDS and IBA results. The dotted diagonal line corresponds to an ideal overlap of EDS and IBA concentrations. Nitrogen concentrations at 2, 4, and 6 sccm N_2_ located above this line suggested +16 at%, +10 at%, and +10 at%, respectively, overestimations of nitrogen concentrations by EDS. Very good agreement between both methods was obtained at 8 sccm and 10 sccm. The absolute deviations of TM concentrations from the ideal line were much smaller, usually below 3 at%. They were clearly influenced by the nitrogen concentration error as a result of the normalization of all element concentrations to 100 at%. At the same time, the good accuracy of the measurement of heavy TM elements by EDS was also confirmed by their overlap with the diagonal in the case of metallic alloy coating.

#### 3.2.3. Phase Composition

The analysis of X-ray diffraction (XRD) measurements was difficult due to the presence of amorphous and nanocrystalline structures in the coatings as well as the lack of diffraction data for the newest multicomponent ceramics in the X-ray diffraction databases. The diffractograms were therefore compared with the results reported for similar transition metal alloy and nitride coatings and bulk systems, as well as for known binary and ternary TM intermetallics and nitrides.

In the metallic TiNbVTaZrHf alloy coating, two to three very wide peaks were present both in B-B and GI configurations (Figure 12a), and they overlapped very well with the earlier azimuthal integral SAED (Figure 4). All three diffractograms implied an amorphous structure similar to that in DCMS HfNbTiVZr coating deposited at room temperature [47] or with a low (16 at%) content of nitrogen [48]. The size of crystallites of only 1 nm, roughly estimated by the Williamson-Hall method, complied with our TEM observations (Figure 4). The amorphous structure, on the other hand, was surprising because a crystalline body-centered cubic (bcc) structure is typical for multicomponent TM alloys [47,48,49,50]. Moreover, substrate temperatures above 275 °C were reported to overcome the effect of suppressed diffusivity and rapid cooling during magnetron deposition, producing and promoting crystallization in these systems [47]. In our case, the absence of crystallization at a 300 °C substrate temperature in our case can be related to the presence of Ta and the difference in power densities between HiTUS and DCMS. The peak positions of the current coating could be compared with those in similar crystalline DCMS TM coatings and with the results of calculations for bcc structures with the lattice parameters a = 0.341–0.342 nm [47,48], 0.336 nm [49], and 0.338 nm [50] and intermetallic ZrTiNb phase (ICDD: 03–065-7192). The approximate peak positions for the fcc structure corresponding to crystalline single and binary nitrides were also added. The experimental peaks did not overlap with the peaks for bcc or fcc structures confirming the amorphous character of the coating.

The diffractograms obtained from highly sub-stoichiometric nitride deposited with 2 sccm N_2_ flow (Figure 12b) were very similar to the HEA case. Only the relative intensity of the main peak in the ≥35° region obtained in B-B was considerably higher than in GI and in the metallic alloy case. Azimuthal integral SAED principally agreed with both X-ray diffractograms, but a shift of the main peak position toward lower 2θ, shoulder above 40°, and additional peaks at 60° and 67° were emphasized. XRD peaks were between the positions of fcc peaks in nitrides (HfTiN_2_, (ICDD:03–065-9258), HfTaN_2_ (ICDD:03–065-9257)_,_ NbTiN_2_ (ICDD:01–089-5134), or TiN (ICDD: 00–038-1420), and NbN (ICDD: 00–038-1155)) and in intermetallics [47,48,49,50]. The size of crystallites obtained by the Williamson-Hall method was around 2 nm, and the microstrains of around 2% corresponded to microstresses over 5.5 GPa. The similarities and differences between the previous coating and GI vs. B-B suggested that the coating structure would be mostly in an amorphous state. The presence of highly localized fcc crystallites seen in Figure 5b,c may not be visible directly by XRD in a much larger coating volume, but it can be anticipated from a narrower and more intense main peak and agreement with the azimuthal integral of SAED. Thus, the structure of this coating can be described as a more pronounced early stage of crystallization and a transition from an amorphous to a textured nanocrystalline structure with crystalline (nano-)columns in nitrogen-rich regions.

In (near-)stoichiometric HT-6TM-6N coating, narrow and intensive XRD peaks typical for crystalline materials appeared, especially in the GI case (Figure 12c. The main GI and B-B peaks at ~41° and ~59° approximately overlapped with the (200) and (220) reflections of fcc nitrides reported in the literature [15,48,51] and with those in TaNbN_2_ (ICDD: 01–089-5133)_,_ HfTiN_2_, TiNbN_2,_ and/or even single nitrides (e.g., TiN, TaN or NbN). The existence of one dominant peak at 41° implied a texture with [100] orientation. Indeed, the texture coefficient for <100> orientation, calculated as a relative intensity of 200 peak to the sum of intensities of all peaks in the B-B diffractogram, was TC_(200)_^BB^ = 0.691, confirming its dominant role. The calculated crystallite size was 10–15 nm. The microstresses estimated from the microstrains of around 0.9% were in the range of 3.2 GPa, which was lower than in the previous sub-stoichiometric coating. The lattice parameter derived from the first five GI peaks was a = 0.4409 nm. All peaks detected in B-B and GI were present in integral SAED, but with many enhanced intensities of 111, 220, 311, and 222 reflections. The peak positions in SAED and XRD were essentially the same.

#### 3.2.4. Mechanical Properties

The results of nanoindentation were summarized as a function of nitrogen flow (Figure 13a) and nitrogen concentration (Figure 13b). Hardness and indentation modulus increased proportionally and almost linearly up to 4 sccm N_2_. Above 4 sccm, stabilization or a small reduction of the obtained values can be seen. Higher substrate temperature in the coating without nitrogen caused a slight increase in the corresponding values, while additional bias caused a slight decrease. However, nitrogen concentration dependences in Figure 13b provided much better physical insight. They reveal a clear maximum at 41.7 at% N_2_ (4 sccm) followed by a mild decrease at 45.5 at% (6 sccm) and abrupt degradation after stoichiometric 50 at% concentration was approached at 8 sccm and 10 sccm N_2_ flows.

The highest hardness of around 33 GPa and indentation modulus of around 400 GPa were achieved in slightly sub-stoichiometric coating with 41.7 at% of nitrogen. The corresponding values in similar slightly sub-stoichiometric 5-TM component nitrides with nitrogen concentrations in the range from 45 at% to 49 at% produced by reactive DCMS were 33–34 GPa and 385–470 GPa, respectively [15,21,48,51]. The overlap with the mechanical properties of the current reactive HiTUS is obvious. Furthermore, a drop in mechanical properties at 50 at% of the nitrogen concentration limit was observed in DCMS coatings [15,48].

The ratio H/E is often used to estimate the toughness of the coatings, assuming 0.1 as a limit for a transition from brittle to ductile behavior [52,53]. The ratio H_IT_/E_IT_ calculated from the values in Figure 13 is shown in Figure 14 as a function of nitrogen content. In metallic coating, the ratio was in the range of 0.085 to 0.090, whereas in the nitride coatings, the majority of the values were even below 0.085, especially in the case of excessive nitrogen flows (10 sccm). The only values exceeding 0.1 were obtained in the coating produced with 2 sccm N_2_. However, it may be an exclusion related to an overestimation of the hardness of this coating (see the deviation from linearity in Figure 13). This assumption was based on the presence of high microstrains and microstresses discussed earlier.

To confirm the low toughness prediction based on the H/E ratio, additional pillar splitting tests were performed on the coating deposited with 6 sccm nitrogen. The basic formula for the calculation is [54]
K_C_ = γ·F_c_/[R^3/2^],
where F_c_ is the critical load when the pillar splits, R is the radius of the pillar, and γ is a dimensionless geometrical factor depending on the coating hardness and elastic moduli of the coating and substrate. The geometrical factor γ ~ 0.231 was taken from FEM results in [55] for H/E =~12 on a stiff substrate (E_substrate_/E_coating_ = 435 GPa/400 GPa = 1.09). The possible difference between cube corner and Berkovich geometry for which the γ was reported was neglected. The tests were performed on four pillars with the diameters R in the range 3.3–3.75 μm using a diamond cube corner indenter. The measured critical loads F_c_ ranged from 8.6 to 12.1 mN. The average fracture toughness value calculated from these four tests was K_C_ = 1.07 ± 0.19 MPa.m^1/2^. Thus, the brittle nature of these coatings implied by their low H/E ratio was confirmed.

## 4. Discussion

At first, reactive HiTUS should be discussed from the viewpoints of hysteresis behavior, target poisoning, and their influence on the deposition rates, stoichiometry, and resulting mechanical properties. As described in Figure 1, Figure 2, Figure 10 and Figure 12, up to 4–6 sccm additions of nitrogen caused a small and approximately linear reduction of ∆*p^av^* and deposition rates accompanied by an almost linear increase of nitrogen content and the level of mechanical properties, respectively. A combination of the considerations of ∆*p^av^* dependence (Figure 1) and nitrogen concentrations suggested that nitrogen was actively consumed because the coatings were highly sub-stoichiometric. It was possible because the amount of sputtered TM available for the reaction with nitrogen was greater than that of nitrogen. Additions of nitrogen in this range into a sputtering atmosphere should increase the working pressure and the deposition rate due to the additional nitrogen atoms. At the same time, increased working pressure should cause a reduction in the deposition rates due to enhanced scattering of sputtered species. The result of these two opposite tendencies is a small decrease in the deposition rates (Figure 2). At 4 sccm N_2_, nitrogen concentration achieved 42 at%, ∆*p^av^* minimum, and mechanical properties their maxima. Above 6 sccm, a transition toward ∆*p^av^* saturation (see Figure 1) due to the gradual consumption of TM supply for the reaction with nitrogen occurred, as indicated by the slow approach of nitrogen concentration to the stoichiometric amount (Figure 10). At 8 sccm and 10 sccm N_2_, an excess of unreacted nitrogen contributed to a substantial increase in working pressure, enhanced scattering, and a significant reduction in the deposition rates by suppressing the ballistic regime of sputtered species transport from the target toward the substrate.

However, the above considerations did not involve the interaction of nitrogen with the unsputtered TM, causing target poisoning. Hysteresis in ∆p^av^ (Figure 1b) was definitely present but it was significantly suppressed. From the viewpoint of reactions between nitrogen and metallic species, it means that the amount of TM available for the reaction included not only sputtered species but also the metal activated by bombardment on the target surface. Such a TM source for nitrogen consumption should be much more powerful than that of sputtered TM in the plasma because of low sputtering yields. At the same time, the active sputtering area in HiTUS is substantially larger than that of the racetrack in conventional DCMS. Assuming the possibility of a gradual increase in nitrogen concentration (stoichiometry) in the film on a very large target surface during poisoning, only slow and gradual changes of corresponding sputtering parameters can be expected in the range of nitrogen additions below the optimum limit. Above that limit, the changes were more pronounced, but apparently only due to the increase in working pressure. The reason is that more nitrogen cannot generate over-stoichiometric poisoning, either on the target or on the substrate. The range of stable deposition regimes during reactive HiTUS would therefore extend well beyond the optimum flow, and the poisoning effects would be significantly suppressed. Thus, target poisoning and subsequent hysteresis behavior in HiTUS play a much smaller role than in conventional reactive magnetron sputtering. Even for multicomponent transition metals systems, the reactive HITUS can be controlled solely by the amount of nitrogen (and possibly by the substrate temperature) without the need for feedback control.

The understanding of the reactive HiTUS process can be used to discuss the mechanisms of structure evolution and corresponding mechanical properties of the deposited coatings. TEM, SAED, and XRD analyses implied that the introduction of a sufficient amount of nitrogen into the solid solution of transition metals resulted in the formation of an fcc structure consisting of [100] and partially [111] textured nano-columns. A stoichiometric fcc nitride solid solution structure with the TM:N ratio of 1:1 would be possible when nitrogen occupies octahedral (Figure 15) interstitial positions (tetragonal positions would correspond to the nitrides with the TM:N ratio of 1:2). It is possible because the ratios R_N_/R_TM_ of atomic radii of nitrogen and TM elements in all binary nitrides (empirical values: R_N_ = 65 pm, R_Ti_ = 140–145 pm, R_Ta_ = 143–145 pm, R_Nb_ = 143–145 pm, R_v_ = 132–135 pm, R_Zr_ = 155–160 pm, R_Hf_ = 155–159 pm [21,56]) were in the range 0.46–0.42. It is substantially smaller than the 0.59 required for solid solutions by geometrical considerations and Hägg’s rule [57].

The lattice parameters of 5-component TM alloys with typically bcc structure were reported in the range from 0.336 nm up to 0.342 nm depending on the presence of individual transition metals [47,48,49,50]. In binary TM nitrides with an fcc structure, they were in the range of 0.441–0.458 nm [51]. In near-stoichiometric multicomponent fcc 5-TM nitrides, lattice parameters were 0.440–0.448 nm in TiZrNbHfV-xN [47], 0.446 nm in TiZrNbHfTa-xN [21], 0.4339 nm in HfNbTiVZr—43 at% N, and 0.4358 nm in over-equimolar Hf-NbTiVZr—43 at% N coatings [51]. Thus, the increase in lattice parameters after the transition from metallic bcc to fcc nitride structure was 28.9–36.3% in binary compounds and 26.8–33.3% in 5-TM-N compounds. In the current HiTUS 6-TM-45.5 at% N coating, the lattice parameter was found to be 0.4409 nm from X-ray diffraction and 0.4496 nm from SAED. The relative difference of around 2% between these values can be related to the presence of local microstrains and composition variations. The relative lattice parameter expansion in comparison with the reference bcc HEA corresponded to 28.9–33.8%. It was in the same range as in the 5-TM-nitride coatings mentioned in [21,47,51].

The introduction of Ta and V into our 6-TM nitrides with near-stoichiometric composition caused only a slight increase of lattice parameters (<2.2% in the case of Ta and 0.8% in the case of V) compared to the reference 5-TM-xN systems. The relative changes in lattice parameters due to nitrogen incorporation were in the range of 30%. Thus, the expansion of interplanar spacings in multicomponent nitrides would result from a combination of a small contribution of a transition from bcc (amorphous in our case) to fcc, the small effect of different TM radii and the dominant contribution of nitrogen concentration.

The stoichiometric multicomponent fcc nitride structure can then be described as a superposition of transition metal fcc sub-lattice and analogous nitrogen sub-lattice shifted by half of the lattice parameter with nitrogen atoms occupying octahedral positions (see Figure 15). This model of stoichiometric fcc nitrides can also be used to describe the formation of sub-stoichiometric nitrides. Since all TM exhibit a strong affinity for nitrogen, the process of gradual filling of the octahedral positions in TM sub-lattice by nitrogen at low nitrogen flows would be controlled only by the amount of nitrogen. It complies with linear relationships between nitrogen flow and its consumption (∆*p^av^* changes) and its concentration in the coatings (see Figure 9). At high nitrogen flows, the transition from linear kinetics toward saturation suggested that the kinetics of nitrogen supply into the few remaining unoccupied interstitials may control the process. The coexistence of small-size crystalline nitride and remaining amorphous metallic phases was in sub-stoichiometric nitride implied from blurred SAED reflections related to nano-crystallites (Figure 5b) and a nearly amorphous structure indicated by X-ray diffraction (Figure 12b). Such a nanocomposite structure would be a natural consequence of local variations in the occupancy of interstitial positions by nitrogen in the case of its deficit at low flows.

A stoichiometric limit of 50 at% clearly seen in Figure 2 (and visible also in the earlier reports [47,48,51]) would be a consequence of a strictly defined number of octahedral interstitial positions. When they are already occupied, excess nitrogen cannot be accommodated in the lattice and must remain in the sputtering atmosphere. The second reason for the existence of a nitrogen consumption limit fits with the earlier discussed amount of sputtered metallic species. At a given voltage (RF energy) applied to the target, it would be given by sputtering yields. That amount of TM (sputtered and on the target surface) would define the “critical flow” of nitrogen available for nitride formation reactions. The analogous mechanism was proposed earlier in W-C:H systems prepared by HiTUS with the involvement of hydrocarbon gases [46]. This approach, combined with target poisoning, explains the discrepancy between the critical relative nitrogen flow of only f_N2_ = 6.25% necessary for stoichiometric composition in the current case (f_N2_ = 8 sccm N_2_/(120 sccm Ar + 8 sccm N_2_) = 6.25%) and f_N2_ = 50% reported in DCMS made coatings [15,48]. The conclusion is that not the relative flow of reactive gas, but the total amount of available TM controls the “stoichiometric” nitrogen flow. Obviously, an increase in the critical flow can be predicted when higher sputtering power is applied.

In near-stoichiometric fcc nitride coatings, preferential growth in [100] and partially in [111] directions were confirmed. Both planes exhibit high packing density, with the (111) plane having energy slightly lower than that of the (100) plane. Because of the small difference in their energies, dominant orientation depends on thickness, competition between interface and strain energies [58,59], as well as the energy of defects involving not only vacancies and dislocations but also nano-twins [60]. The full width at half maxima (FWHM) of 200 peak in near-stoichiometric 6-TM-6N coating (Figure 12c) was Γ_200_^BB^ = 1.24°. It was within the range Γ_200_^BB^ = 1.34°–1.19° reported for similar DCMS TiTaVZrHf-N coatings deposited with 30%, 40%, and 45% relative N_2_ flows [15], but substantially higher than Γ_200_^BB^ = 0.76–0.89 in near-stoichiometric TiN coatings [61]. FWHM is primarily directly proportional to defect concentration and microstresses and inversely proportional to crystallite domain size [15]. When relatively small crystallite size and microstresses were taken into account, high FWHM indirectly implied high defect concentrations. Thus, the occurrence of both preferred orientations and even transitions between them during coating growth could be expected, especially in highly defective sub-stoichiometric nitrides (Figure 5c).

Hardness and indentation modulus dependences principally followed the above model of nitride formation based on ∆*p^av^*, deposition rates, and nitrogen concentration dependences. It suggested that the increase of nitrogen concentration directly correlated with the mechanical properties up to a certain limit. Degradation of properties above critical flow can be related to lower densities of the coating due to the loss of energy of impinging species resulting from enhanced scattering of sputtered species in the plasma at increased working pressures. However, the question remains if this argument is sufficient to explain why the highest mechanical properties were achieved in slightly sub-stoichiometric (42 at% of nitrogen) and not in the stoichiometric coatings. Another remaining question is associated with the brittle behavior of the current amorphous 6-TM-HEA coating. Its H_IT_/E_IT_ ratio was 0.086, which was above 0.05–0.078 reported in similar but crystalline 5-component DCMS HfNbTiVZr coatings [15,47,48,51]. It was found, that although high quenching rates during magnetron sputtering on room temperature substrates favored single-phase crystalline bcc solid solutions, at higher (450 °C) substrate temperatures, dual bcc and Laves phases may appear [47]. Low H_IT_/E_IT_ ratio and toughness were related to the presence of brittle Laves phases formed in the bulk at temperatures below 800 °C [50]. Contrary to these conclusions, the current HiTUS 6-TM alloy coating deposited at 300 °C was amorphous, the Laves phase was not detected, and its H_IT_/E_IT_ ratio was higher. Obviously, additional investigations are necessary to clarify the reasons for low toughness and to explore possible toughening mechanisms in the multicomponent TM alloy and nitride coatings.

Finally, possibly the most important output of the current work for industrial applications is the demonstration of the ability of reactive HiTUS to produce multicomponent coatings with mechanical properties equal to those in a similar arc and DCMS metallic and nitride coating systems [9,10,12,15,16,17,18,19,20,21,22,23,24,25,47,48,49,50,51] without a feedback control of target poisoning. The structures of the studied coatings, despite variations and deviations from equiatomic concentrations of TM elements at different scales, corresponded to homogeneous solid solutions regardless of the amount of nitrogen. Thus, the studied HiTUS 6 TM metallic alloy and nitride coatings may be attributed to the materials with TM sub-lattice stabilized by high configurational entropy [15], sometimes called “high entropy nitrides”. The equiatomic (or equimolar) concentrations of all transition metal elements required in high entropy alloys by definition [3,4] do not seem to be strict conditions. Indeed, solid solutions of metallic sub-lattice remained homogeneous and exhibited very high mechanical properties even in the coatings with substantial deviations of individual TM from ideal concentration.

## 5. Conclusions

The investigations of the pressure changes during reactive HiTUS, structure, chemical, and phase composition, and their correlations with the resulting mechanical properties of TiNbVTaZrHf-xN coatings, led to the following conclusions:Formation of sub- and near-stoichiometric nitride structure from high entropy stabilized metallic system during reactive HiTUS can be described by a gradual increase of the occupancy of octahedral interstitial position in the face-centered cubic lattice by nitrogen atoms;The highest values of hardness H_IT_ ~ 33 GPa and indentation modulus E_IT_ ~ 400 GPa were achieved in slightly sub-stoichiometric (~42 at% nitrogen) coatings;Hysteresis behavior in reactive HiTUS is significantly suppressed. The structure and mechanical properties of the studied TiNbVTaZrHf-xN coatings can be controlled only by the amount of reactive nitrogen without a need for feedback control;HiTUS and reactive HiTUS are suitable for the deposition of 6-TM alloy and nitride coatings characterized as homogeneous solid solutions with metallic sub-lattice stabilized by high configurational entropy.

## Figures and Tables

**Figure 1 materials-16-00563-f001:**
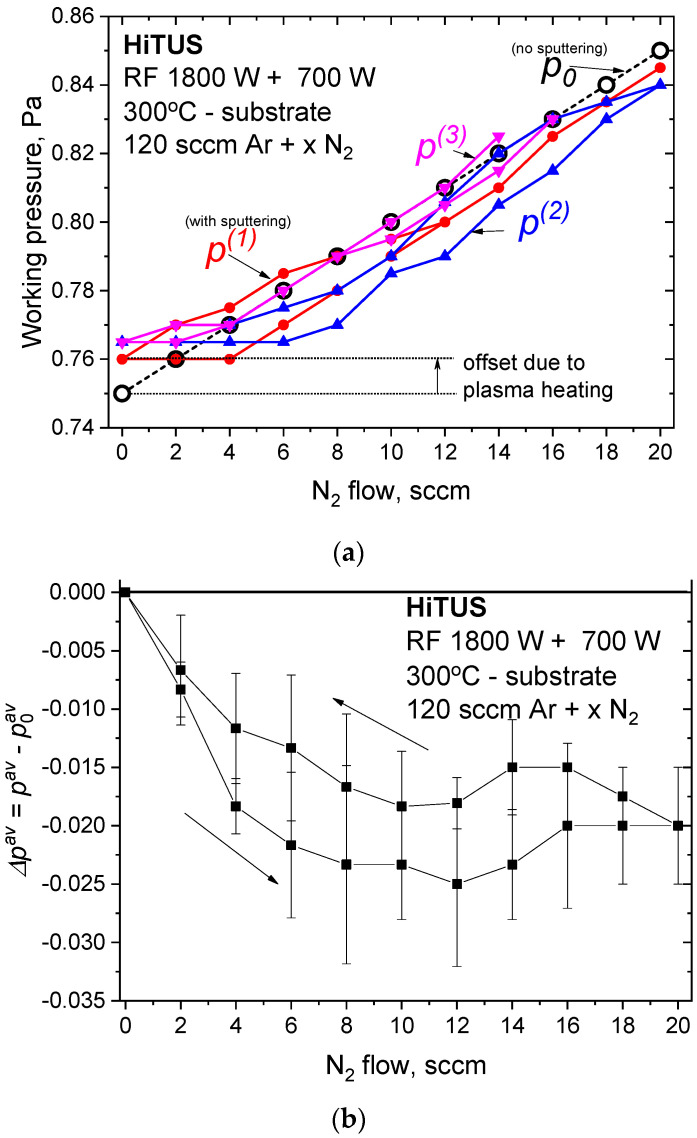
(**a**) Changes in working pressure in the deposition chamber due to nitrogen additions into the Ar sputtering atmosphere without (*p*_0_) and with the power of RF 700 W applied to TiNbVTaZrHf target during three independent nitrogen flow cycles (*p*^(1)^, *p*^(2)^, *p*^(3)^) (the offset of pressure at 0 sccm N_2_ after turning sputtering on is an artifact attributed to plasma heating); (**b**) average pressure difference ∆*p^av^ = p^av^ − p*_0_ resulting from nitrogen consumption.

**Figure 2 materials-16-00563-f002:**
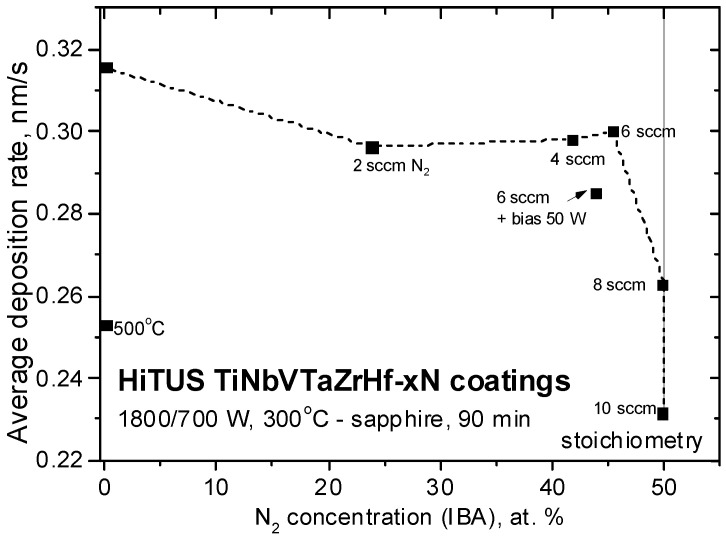
The average deposition rates of the studied HiTUS TiNbVTaZrHf-xN coatings at different nitrogen flows as a function of nitrogen atomic concentrations (nitrogen concentrations were taken from later IBA measurements).

**Figure 3 materials-16-00563-f003:**
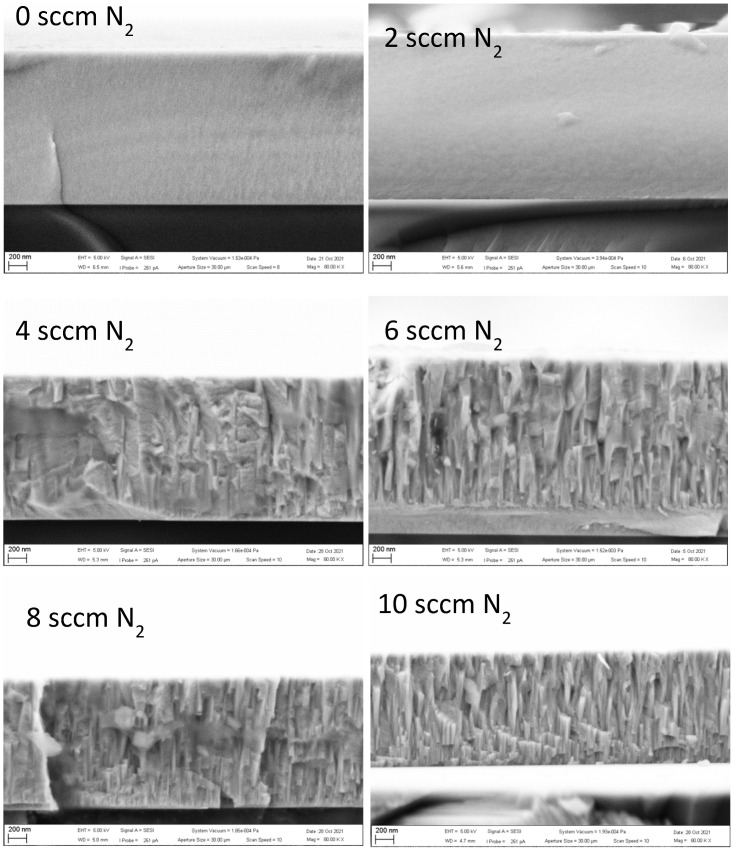
Fractured cross-sections of HiTUS TiNbVTaZrHf-xN coatings deposited with different flows of N_2_ in Ar sputtering atmosphere.

**Figure 4 materials-16-00563-f004:**
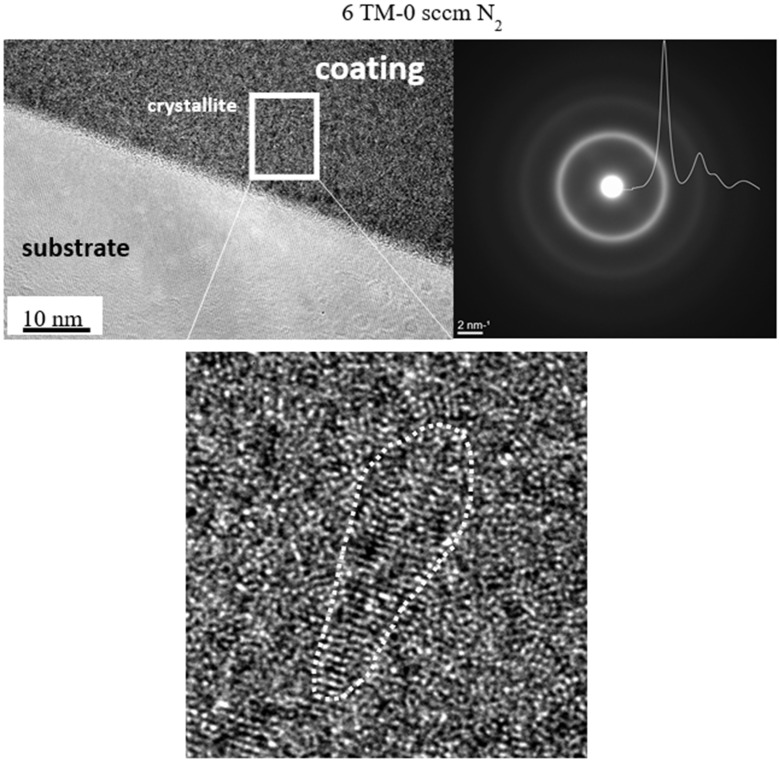
Structure of HiTUS TiNbVTaZrHf coatings deposited without nitrogen additions. SAED spectrum indicates an amorphous structure despite the occasional presence of small crystallites detected by HRTEM (see insert). Azimuthal integral SAED also corresponds to an amorphous structure.

**Figure 5 materials-16-00563-f005:**
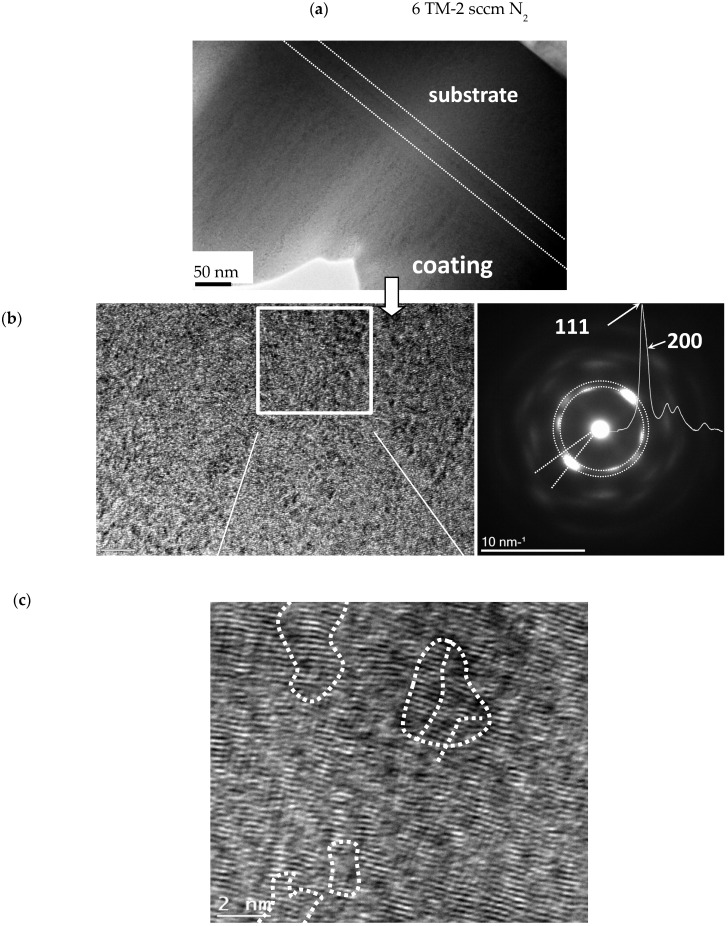
Structure of the reactive HiTUS TiNbVTaZrHf-xN coating deposited with 2 sccm N_2_: (**a**) low magnification TEM image; (**b**) HRTEM images and the corresponding SAED with azimuthal integral SAED; (**c**) Fast Fourier transformation (FFT) enhanced cut-off with possible boundaries between selected and highly defect crystalline domains.

**Figure 6 materials-16-00563-f006:**
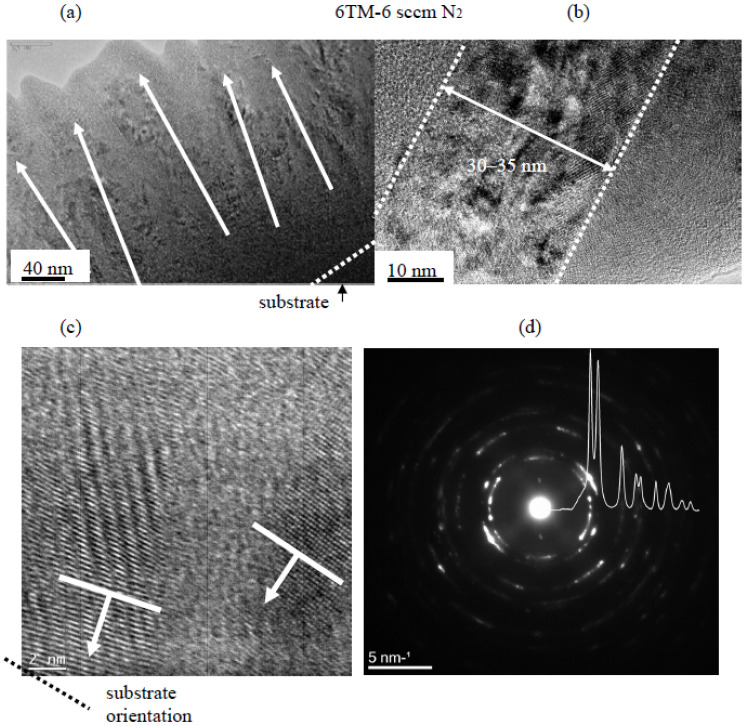
TEM structure of reactive HiTUS TiNbVTaZrHf-6N coating deposited with 6 sccm N_2_: (**a**) low magnification image demonstrating slightly disoriented columnar grains approximately perpendicular to the substrate embedded in the amorphous matrix phase; (**b**) HRTEM of the columnar grain and (**c**) FFT enhanced image of the columnar grains and (**d**) corresponding SAED and its azimuthal integral.

**Figure 7 materials-16-00563-f007:**
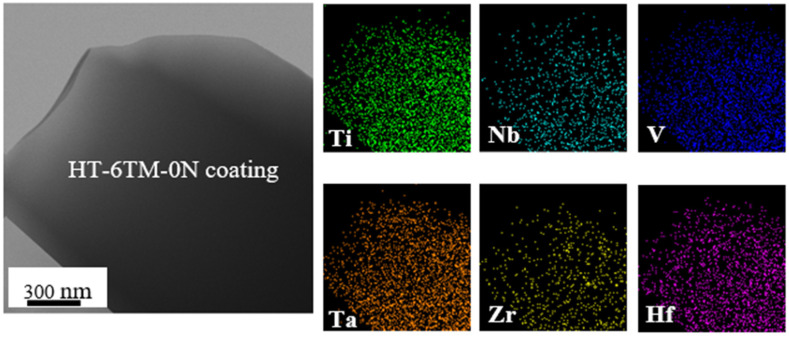
EDS mapping of the distribution of transition metal elements in the studied HiTUS–TiNbVTaZrHf-0N coating (~1705 nm thick).

**Figure 8 materials-16-00563-f008:**
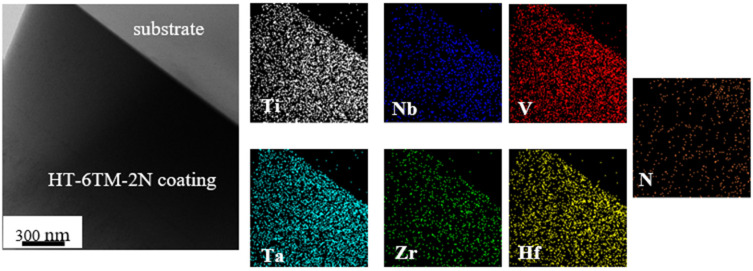
The distribution of nitrogen and TM elements in TiNbVTaZrHf-2N coating by EDS.

**Figure 9 materials-16-00563-f009:**
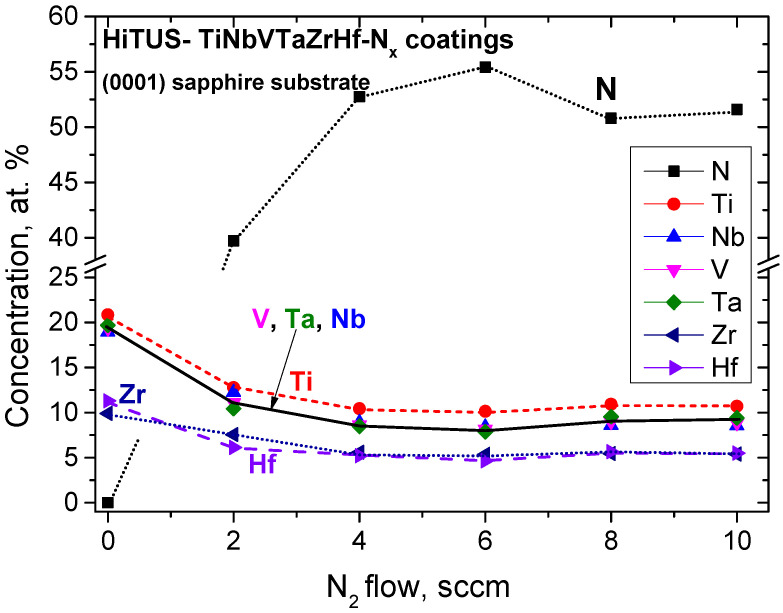
Average semi-quantitative EDS compositions of the studied reactive HiTUS TiNbVTaZrHf-xN coatings (x–flow of nitrogen in sccm) as a function of nitrogen additions in the Ar sputtering atmosphere. The averaging was based on top-view measurements in SEM from four large (900 × 680 µm^2^) areas.

**Figure 10 materials-16-00563-f010:**
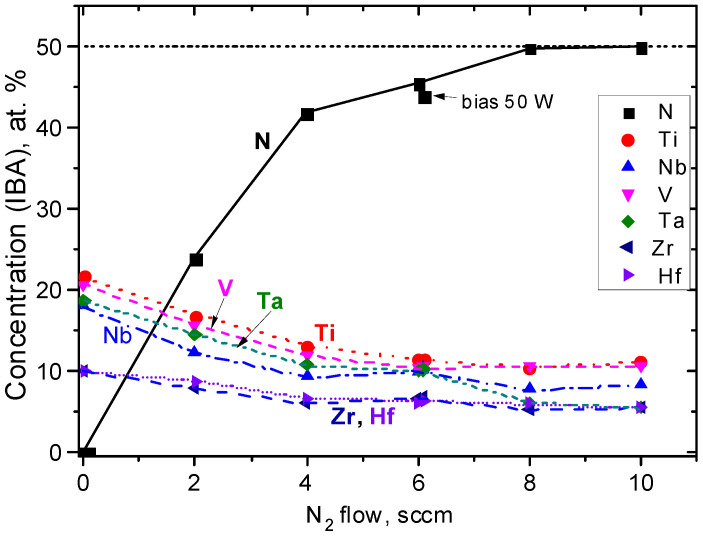
The concentrations of TM and nitrogen determined by IBA methods in the studied reactive HiTUS TiNbVTaZrHf-xN coatings in dependence on the additions of nitrogen.

**Figure 11 materials-16-00563-f011:**
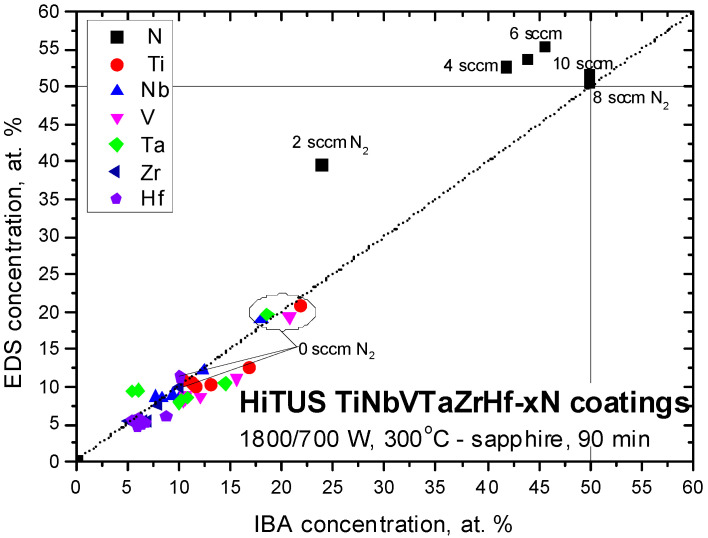
The comparison of the concentrations of TM and nitrogen in the studied reactive HiTUS TiNbVTaZrHf-xN coatings obtained by EDS/SEM and IBA methods. The dotted diagonal line indicates an ideal overlap between both methods.

**Figure 12 materials-16-00563-f012:**
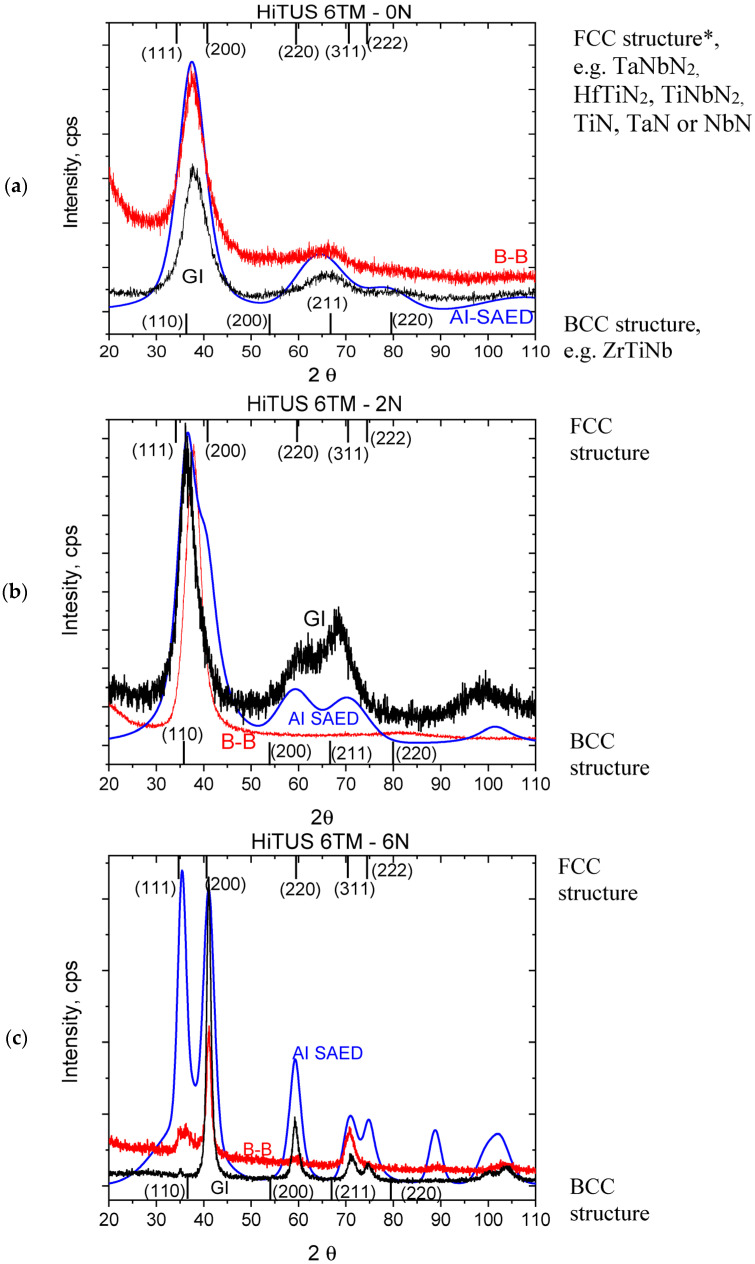
X-ray diffraction patterns of the studied reactive HiTUS TiNbVTaZrHf-xN (HT-6TM-xN) coatings in symmetrical θ/2θ (Brag–Brentano (B-B)), grazing incidence (GI), and azimuthal integral of SAED (AI SAED) from TEM observations and prepared: (**a**)—without nitrogen addition (0 sccm N_2_); (**b**)—with 2 sccm N_2_ and (**c**)—6 sccm N_2_. (*—from [47,48,49,50]).

**Figure 13 materials-16-00563-f013:**
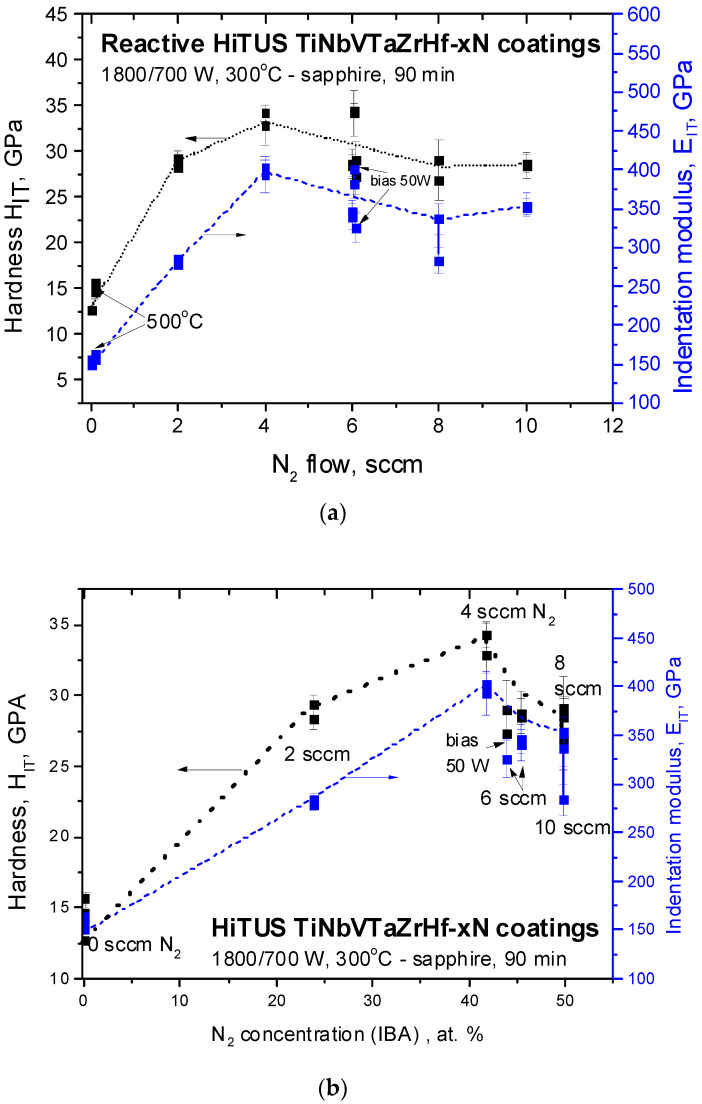
The dependences of hardness and indentation modulus in the studied reactive HiTUS TiNbVTaZrHf-xN coatings on (**a**) nitrogen flow in Ar sputtering atmosphere and (**b**) IBA nitrogen concentrations.

**Figure 14 materials-16-00563-f014:**
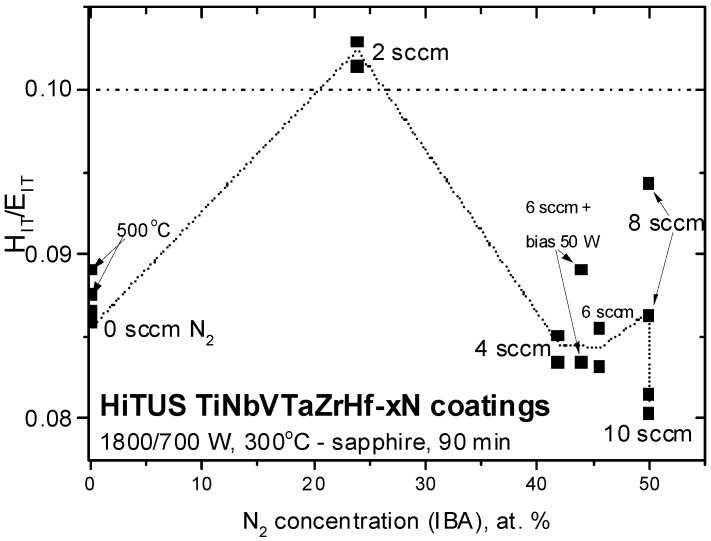
The dependence of H_IT_/E_IT_ ratio on nitrogen concentration in the studied reactive HiTUS TiNbVTaZrHf-xN coatings.

**Figure 15 materials-16-00563-f015:**
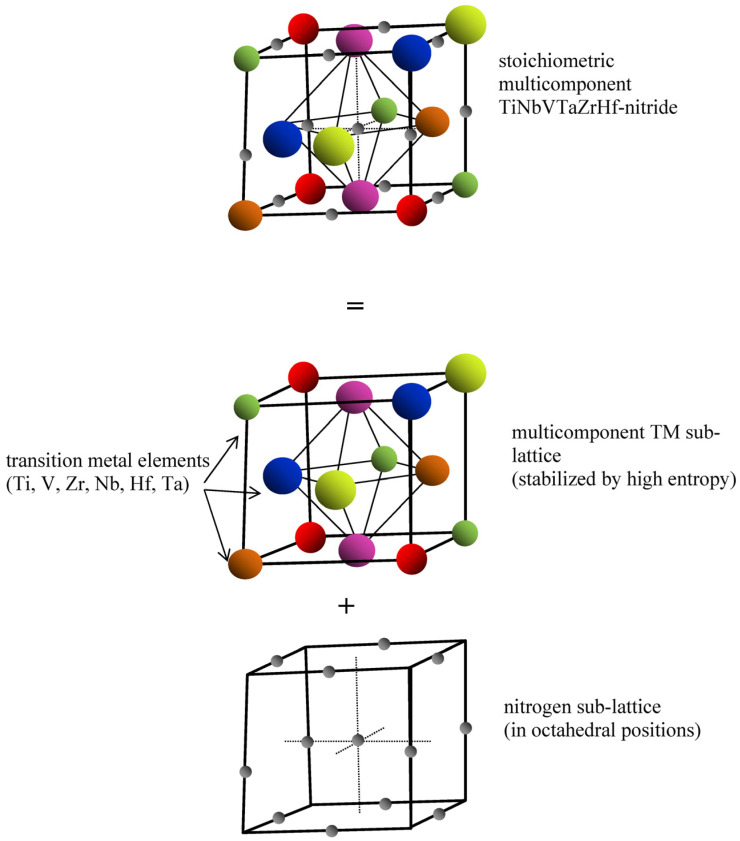
Schematic unit cell of stoichiometric multicomponent fcc nitride with nitrogen in octahedral interstitial locations described as a superposition of fcc TM and nitrogen sub-lattices.

**Table 1 materials-16-00563-t001:** The nominal and true average compositions of the target obtained by averaging 7 EDS/SEM (top view) measurements from the surface area of 900 × 680 µm^2^.

Target	Ti, at%	Nb, at%	V, at%	Ta, at%	Zr, at%	Hf, at%
Nominal composition	20	18	20	18	12	12
EDS * composition (after sputtering without nitrogen)	18.5 ± 1.9	19.6 ± 1.3	26.4 ± 3.6	13.0 ± 0.4	10.4 ± 0.3	12.1 ± 0.7

* This standard deviation refers to the scatter of calculated concentrations across seven different sites, not to the precision of EDS concentration measurement.

**Table 2 materials-16-00563-t002:** The list of the studied TiNbVTaZrHf–N coatings prepared by reactive HiTUS and their thicknesses measured on fracture cross-sections (see Figure 3).

Sample	N_2_ Flow sccm	Thickness nm
HT-6TM-0N	0	1705
HT-6TM-0N-500 °C	0	1370
HT-6TM-2N	2	1600
HT-6TM-4N	4	1610
HT-6TM-6N	6	1620
HT-6TM-6N-50Wbias	6	1540
HT-6TM-8N	8	1420
HT-6TM-10N	10	1250

**Table 3 materials-16-00563-t003:** Average concentrations of transition metal elements in the target compared to those in the HT-6TM-0N coating obtained on thin foil by EDS/TEM and in top view EDS/SEM from a large area (900 × 680 µm^2^).

TM	EDS/SEM Concentration in the Target at% *	EDS/TEM Concentration at% (Thin Foil) *	EDS/SEM Concentration at% (Top View) *
Ti	18.5 ± 1.9	32.1 ± 4.7	20.9 ± 0.2
Nb	19.6 ± 1.3	17.2 ± 3.6	19.0 ± 0.2
V	26.4 ± 3.6	22.8 ± 3.3	19.3 ± 0.1
Ta	13.0 ± 0.4	14.2 ± 2.4	19.7 ± 0.2
Zr	10.4 ± 0.3	8.3 ± 1.8	9.9 ± 0.1
Hf	12.1 ± 0.7	5.5 ± 0.7	11.3 ± 0.1

* This standard deviation refers to the scatter of calculated concentrations across four different sites, not to the precision of EDS concentration measurement.

**Table 4 materials-16-00563-t004:** Local and average concentrations of each element in HT-6TM-6N coating obtained by EDS in three measurement configurations: on TEM foil, on fracture surface in SEM, and in the top view from four large areas (900 × 680 µm^2^) in SEM.

Element	N, at%	Ti, at%	Nb, at%	V, at%	Ta, at%	Zr, at%	Hf, at%
Area 1 TEM	50.9	16.4	8.7	8.9	6.8	5.9	2.3
Area 2 TEM	43.1	19.7	9.4	12.2	6.4	6.5	2.8
Area 3 TEM	42.5	16.1	12.6	11.9	7.3	6.9	2.9
Area 4 TEM	55.4	14.0	10.5	8.1	5.3	5.0	1.9
Mean ± standard deviation *	48.0 ± 5.4	16.6 ± 2.0	10.3 ± 1.5	10.3 ± 1.8	6.4 ± 0.7	6.0 ± 0.7	2.5 ± 0.4
SEM (fracture surface)	48.0	11.9	8.0	9.4	10.7	5.1	6.3
SEM (top-view) mean ± standard deviation *	**55.4 ± 0.5**	**10.1 ± 0.2**	**8.4 ± 0.1**	**8.2 ± 0.1**	**7.9 ± 0.2**	**5.3 ± 0.1**	**4.7 ± 0.1**

* This standard deviation refers to the scatter of calculated concentrations across four different sites, not to the precision of EDS concentration measurement.

**Table 5 materials-16-00563-t005:** The comparison of the concentrations of transition metals and nitrogen in the studied HiTUS TiNbVTaZrHf-xN coatings determined by EDS/SEM and by IBA methods.

N_2_ Flow Sccm	N EDS at%	N IBA at%	Ti EDS at%	Ti IBA at%	Nb EDS at%	Nb IBA at%	V EDS at%	V IBA at%	Ta EDS at%	Ta IBA at%	Zr EDS at%	Zr IBA at%	Hf EDS at%	Hf IBA at%
0	0	**0**	20.9	**21.7**	18.9	**18.0**	19.4	**20.7**	19.7	**18.6**	9.9	**10.1**	11.3	**10.1**
0 500 °C	0	**-**	20.7	**-**	16.1	**-**	21.4	**-**	18.3	**-**	8.8	**-**	14.7	**-**
2	39.7	**23.8**	12.8	**16.8**	12.3	**12.4**	11.1	**15.7**	10.4	**14.5**	7.6	**8.0**	6.1	**8.8**
4	52.8	**41.7**	10.4	**13.1**	8.9	**9.4**	8.6	**12.0**	8.5	**10.9**	5.6	**6.2**	5.2	**6.7**
6	55.4	**45.5**	10.1	**11.5**	8.4	**10.0**	8.2	**10.5**	7.9	**10.0**	5.3	**6.5**	4.7	**6.0**
6 bias	53.8	**43.8**	10.4	**11.4**	8.5	**10.3**	8.5	**10.7**	8.3	**10.5**	5.4	**6.9**	5.0	**6.4**
8	50.8	**49.8**	10.9	**10.4**	8.6	**7.8**	9.0	**10.6**	9.6	**6.1**	5.5	**5.2**	5.7	**6.1**
10	51.6	**49.9**	10.7	**11.1**	8.5	**8.3**	8.9	**10.7**	9.4	**5.5**	5.4	**5.5**	5.5	**5.5**

## Data Availability

The data presented in this study are available on request from the corresponding author.

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
