# Peer review of "Reactive HiTUS TiNbVTaZrHf-Nx Coatings: Structure, Composition and Mechanical Properties"

_materials, 2023, doi:10.3390/ma16020563_

Round 1
Reviewer 1 Report
In this paper, TiNbVTaZrHf xN coatings were prepared by HiTUS technology, and the hysteresis behavior in reactive sputtering process, as well as the structure, composition, composition and mechanical properties of the obtained coatings were studied. The following issues need to be clarified:
1 "TiNbVTaZrHf bond layers up to 200 nm thick were applied, with the gradient composition result from 1 sccm step increases in nitrogen flow from 0 sccm up to a given flow." How can this be reflected in Figure 3?
2 In Fig. 12, the author gives the XRD spectrum of the sample. In combination with Fig. 12, the high-resolution transmission pictures and electron diffraction patterns in Fig. 5 and Fig. 6 should be comprehensively phase analyzed.
3 The coating prepared by sputtering usually has a certain stress. When analyzing its mechanical properties such as hardness and elastic modulus, does the author consider the influence factors of stress? It is suggested to supplement the stress data of different samples and analyze their corresponding relationships.
4 Figure 15 shall be supplemented with corresponding legend.
Author Response
1 "TiNbVTaZrHf bond layers up to 200 nm thick were applied, with the gradient composition result from 1 sccm step increases in nitrogen flow from 0 sccm up to a given flow." How can this be reflected in Figure 3?
The introduction of graded bond layer was an undesired complication for us (besides problems with hardness measurements), however, it had to be used to avoid strong delamination occurring in the coatings with relatively small nitrogen flows (at 2 and 4 sccm). The understanding is that high residual stresses were generated in sub-stoichiometric nitrides due to insufficient occupancy of interstitials by nitrogen and high deformation of the lattice (it was confirmed by high microstrains). In metallic coatings and in near-stoichiometric nitrides, the stresses were lower and there was no problem with delamination. However, to keep the conditions uniform for the whole series, the bond layers were applied to all coatings.
Obviously, the thickness of the bond layer depended on the flow of nitrogen used for the top coating. It was zero in the coating without nitrogen and therefore, cannot be seen in Fig. 3. In the coating with 2 sccm N2, it consisted of pure metal deposited for 1 min and another deposition for 1 min with 1 sccm N2 followed by regular top coating with 2 sccm N2. Because of more 1 sccm nitrogen deposition steps in graded interlayer under the top coatings with higher nitrogen additions, the maximum thicknesses up to around 200 nm (around 20 nm/min was added in each step) can be expected.
The visibility of bond layer and coating structure on fracture surfaces was affected by several factors. Possibly the most important was the way it was fractured. In brittle materials, the distance from the initiation center of fracture affects the topography of the fracture surface (mirror zone always looks amorphous and hackle zone always “crystalline” regardless of the structure). Then, erroneous conclusions concerning coating structure (and bond layer) can be made based on fracture surface appearance. Second factor is a difference in composition between bond layer and top coating. It was invisible when the difference was small (Fig. 3 – at 0, 2, 4 sccm). Final factor was the structure. Initial metallic coating was amorphous but nitrogen addition caused formation of textured fcc structure. When oriented crystalline grains formed already in the later sublayers of the graded bond layer, they could continuously grow into top layer. The boundary between bond layer and top layer would be therefore fuzzy and hardly can be determined from SEM micrographs even at 50k magnifications.
The above mentioned factors cast more questions than benefits on the results which can be obtained from coating fracture surfaces. Therefore, SEM observations in Fig. 3 were used only for average deposition rate calculation and the information about coating structures was considered only as complimentary to TEM observations. The discussion on the visibility of interlayers based on the above considerations would result in further expansion of the manuscript without true benefits. However, to address the issues in the reviewer’s case, the text in the paragraph related to interlayer was slightly modified. The current version is
The substrates involved polished (0001) sapphire and (111) Si wafers, which were ultrasonically cleaned in acetone and by plasma just prior to the onset of deposition. To ensure good coating adhesion, gradient TiNbVTaZrHf-xN bond layers were applied. Gradient composition was obtained by gradual increase of nitrogen flow additions into the sputtering atmosphere with 1 sccm and 1 min step starting from x = 0 sccm up to a nitrogen flow in top coating. Based on the deposition rate of ~20 nm/min, the bond layer thicknesses varied from 0 nm up to around 200 nm in the coatings produced with the highest nitrogen additions. Despite the importance of bond layers for coating adhesion, their role in structure formation and mechanical properties was not considered.
2 In Fig. 12, the author gives the XRD spectrum of the sample. In combination with Fig. 12, the high-resolution transmission pictures and electron diffraction patterns in Fig. 5 and Fig. 6 should be comprehensively phase analyzed.
Fig. 12 combined the results from SAED and XRD to emphasize the similarities and differences between highly localized TEM and XRD measurements averaged over relatively large area. Moreover, similar differences were explored using EDS (localized TEM and averaged SEM) vs. ERDA measurements of chemical compositions. The description of only objective results in the Results part and subsequent subjective interpretation of these results in Discussion was preferred prior to more extensive description of the relations between AI SAED and XRD diffractograms in Fig. 12 in Results. The analysis and complementarity of TEM and XRD results in the Discussion seems to be extensive and adding more details in the Results part not necessary. Moreover, the work covers topics starting from reactive plasma deposition up to final mechanical properties of coatings. Despite there is always a possibility for more comprehensive analysis of TEM results, it would require additional observations and most probably, another manuscript. The expansion into TEM analysis would be beyond the scope of the work (and our current abilities).
3 The coating prepared by sputtering usually has a certain stress. When analyzing its mechanical properties such as hardness and elastic modulus, does the author consider the influence factors of stress? It is suggested to supplement the stress data of different samples and analyze their corresponding relationships.
We fully agree with this remark, residual stresses were present as mentioned in the answer concerning bond layer and in the work based on the measurement of microstrains. However, the consideration of the residual stresses in nanoindentation measurements would require additional measurements in unstressed coatings which were not available. Moreover, the measurements of microstrains using XRD were not accurate enough due to wide peaks in amorphous/nanocomposite coating structures. Thus, not only the effects of bond layer but also those of residual stresses on nanoindentation measurements were omitted. To emphasize this fact, a sentence explaining it was added at the end of the Experimental part:
The effects of bond layer and possible residual stresses on indentation hardness and modulus values were not considered.
4 Figure 15 shall be supplemented with corresponding legend.
The legend is present in Fig. 15. This remarks is possibly a misunderstanding resulting from a shift of the caption in Fig. 15 on the next page.
Reviewer 2 Report
The manuscript can be accepted in present form.
The manuscript “Reactive HiTUS TiNbVTaZrHf-Nx coatings: structure, composition and mechanical properties” deals with the issue of reactive sputtering, namely the elimination of hysteresis effect. Reactive sputtering is used to obtain High Entropy Alloys, a class of materials that are very interesting for applications and for their special properties. Moreover the technique employed uses high target utilization, a key aspect when dealing with industrial applications. The manuscript is carefully written, providing all the information needed to comprehend the process, on one side, and the materials properties , on the other side. Form this point of view, it is interesting both form the fundamental perspective, of understanding the main processes, and for the applications. Complementary techniques are used when needed, i.e RBS and EDS, to obtain all the relevant information.
The conclusion summarize the main findings of the manuscript.
Author Response
No changes in the manuscript were required
Reviewer 3 Report
The work by Lojaf et al concerns the deposition of a 6-component transition metal nitride (TiNbVTaZrHf-Nx) using the process of HiTUS. The authors provide a wide-range characterization of the coatings produced with different N content. Compositional, structural, morphological, and mechanical aspects are investigated along with the hysteresis behaviour of the process using HiTUS. The results are compared with DCMS and indicate the advantage of the novel process in terms of hysteresis suppression and control, while yielding coatings with comparable properties. The deposition of the chosen alloy and the evolution of its structure with N content is also novel. The relevance of the work is clear, and the text is well-written and of interest to the potential readers. Therefore, I suggest the publication of this manuscript after the consideration of some points:
1. What is the voltage correspondent to the 5 W and 50 W bias applied to the substrates? Reporting it in terms of tension makes the comparison with literature more straightforward.
2. What is the motivation for using two different substrates?
3. In Fig. 4a, the visualization of the small crystallites is quite difficult. Providing a FFT enhanced image of the crystallite region (as is presented in Figs. 5c and 6c) could help.
4. “The increase of substrate temperature to 500°C in the coating without nitrogen and the bias in the case with 6 sccm N2 resulted in a decrease in deposition rates. The reasons may be related to enhanced scattering at higher atmosphere and substrate temperature and re-sputtering from the coatings due to a higher bias.”. The change in deposition rate at 500°C is quite large. How are the substrates heated? Is it localized heating or the whole chamber is heated? Could you report the deposition pressure at 500°C? It would help verify the scattering explanation.
5. The authors should consider moving some of the chemical composition results to a Supplementary Material section. For example, there is no need to present the values of four different EDS measurements in the main text, as is done in Table 4. Only the average and standard deviation are sufficient. Also, the comparison between top-view and cross-sectional SEM EDS do not bring any relevant aspect to the discussion and could also be presented in the Supplementary Material section.
6. The deviations between different chemical composition measurements would be best reported in the same table where the results for each technique are listed. Writing it in text form is confusing, especially when there are so many different measurements involved.
7. What can be discussed about the samples produced with higher bias and higher substrate temperature in terms of structure and mechanical properties?
Author Response
- What is the voltage correspondent to the 5 W and 50 W bias applied to the substrates? Reporting it in terms of tension makes the comparison with literature more straightforward.
We agree with the remark. The corresponding DC voltage was not measured directly but based on the calibration curve from the producer, RF 5 W corresponds to approximately -45 V and 50 W to around -295 V. Our previous experience with DC bias was that its application is beneficial up to around -50 V, higher tensions caused additional problems. Obviously, RF 50 W was well above the recommended range as it was confirmed experimentally. The explanations showing corresponding DC voltage in RF bias were added in the Experimental part and in the Results. Additional explanation was also provided at the end of chapter Deposition and deposition rates (all these changes are marked by blue background (see also the answer No. 7).
- What is the motivation for using two different substrates?
Multicomponent TM nitrides are expected to exhibit high thermal resistance therefore, sapphire substrates were used. To preserve sapphire substrates intact for thermal and other tests, additional coatings were simultaneously deposited on Si substrates. These samples were used for coating thickness measurements. The sentence
The coatings deposited on Si substrates were used for thickness measurements on the fractured cross sections.
was added into the Experimental part to address this remark.
Another reason was that nanoindentation measurements were performed on the coatings deposited simultaneously on both substrates to investigate the effect of substrate stiffness on the calculated values. However, this study is beyond the scope of the current work.
- In Fig. 4a, the visualization of the small crystallites is quite difficult. Providing a FFT enhanced image of the crystallite region (as is presented in Figs. 5c and 6c) could help.
FFT enhanced image is right now not available (Christmas holiday…) therefore, a magnified detail was added as an insert in Fig. 4.
- “The increase of substrate temperature to 500°C in the coating without nitrogen and the bias in the case with 6 sccm N2 resulted in a decrease in deposition rates. The reasons may be related to enhanced scattering at higher atmosphere and substrate temperature and re-sputtering from the coatings due to a higher bias.”. The change in deposition rate at 500°C is quite large. How are the substrates heated? Is it localized heating or the whole chamber is heated? Could you report the deposition pressure at 500°C? It would help verify the scattering explanation.
Substrate temperature is definitely a parameter affecting coating structure. In the work of Fritze et al. [see ref. 51 in the manuscript] on similar DCMS nitride coatings, (crystalline) bcc structure was obtained in metallic coating already at 275oC and even Laves phase identified at 450oC while coating remained amorphous only at RT. In our HiTUS metallic coating, amorphous structure occurred even at 300oC (substrate was heated using conventional resistive furnace located just above the substrate). That was the reason additional metallic coating was produced at 500oC. Despite there were slightly more pronounced peaks in the diffractogram, the increase of substrate temperature resulted in a decrease of deposition rate and only marginal increase of mechanical properties. Therefore, substrate temperature of 300oC was used for the whole series.
I am not sure if further explanation in the text of the work would be beneficial without addition of a diffractogram from the coating at 500oC and further discussion. Thus, no additional explanation was made in the text.
- The authors should consider moving some of the chemical composition results to a Supplementary Material section. For example, there is no need to present the values of four different EDS measurements in the main text, as is done in Table 4. Only the average and standard deviation are sufficient. Also, the comparison between top-view and cross-sectional SEM EDS do not bring any relevant aspect to the discussion and could also be presented in the Supplementary Material section.
Although we principally agree with this remark, the existing arrangement is preferred because of the emphasis on the variability of very local compositions on one hand and limited accuracy of quantitative EDS and IBA measurements on the other hand. Neither HEA nor HEN are homogeneous at nano-scale, the variability of individual TM is w ithin 2-3 at % and that of nitrogen within 10 at%. When these data are moved into Supplement, direct comparison of the local variations and average macroscale concentration values would be partially lost. The benefit resulting from the shortening of the table and text seems to be smaller.
Additional, though less important reason for indicating several local values in Tab. 4 is that in many papers, the concentrations of elements obtained by EDS are reported with two decimal digits. It is obvious that such accuracy is not real.
- The deviations between different chemical composition measurements would be best reported in the same table where the results for each technique are listed. Writing it in text form is confusing, especially when there are so many different measurements involved.
I am not sure what is the meaning of the suggestion. Tab. 3 is related only to the target and metallic coating, Tab. 4 only to nitride coating produced with 6 sccm N2 in different EDS measurement modes. Their unification into one table would be confusing because of mixing metallic and nitride coatings. Tab. 5 relates to the “best” (large scale and top view) average values obtained by EDS compared with IBA results on all coatings as a function of nitrogen flows. They also could not be united with the previous data without confusion. Tab. 5 could be principally deleted leaving just Figs. 9-11 having the same content but shortening would be not significant and direct comparison of numerical values would be more difficult. The description of Tab 5 and Figs 10 and Fig. 11 concentrated only on the main differences of IBA compared to EDS measurements to emphasize semi-quantitative characterization by EDS. Shortening or removal of the last two paragraphs in Structure and chemical composition chapter would be not beneficial. The text was therefore left as it was.
- What can be discussed about the samples produced with higher bias and higher substrate temperature in terms of structure and mechanical properties?
This question was already partially addressed in the answer No. 4. Substrate temperature 500oC slightly promoted crystallization compared to 300oC but the temperature (and/or time of deposition) was not sufficient for more pronounced crystallization. RF bias corresponding to almost -300 V DC caused reduction of deposition rates attributed to re-sputtering and without substantial effect on mechanical properties. The final paragraph in the chapter Deposition and deposition rates was extended and it includes:
The reasons may be related to enhanced scattering of sputtered species in Ar atmosphere at higher temperatures. RF bias of 50 W corresponding to almost -300 V in DC case was comparable to the values applied to the target during DC sputtering. Thus, the reduction of the deposition rate could be attributed to re-sputtering from the growing coating.